# Pharmacological Treatment of Ascites: Challenges and Controversies

**DOI:** 10.3390/ph18030339

**Published:** 2025-02-27

**Authors:** Jimmy Che-To Lai, Junlong Dai, Lilian Yan Liang, Grace Lai-Hung Wong, Vincent Wai-Sun Wong, Terry Cheuk-Fung Yip

**Affiliations:** 1Medical Data Analytics Center, Department of Medicine and Therapeutics, The Chinese University of Hong Kong, Hong Kong SAR, China; jimmyctlai@cuhk.edu.hk (J.C.-T.L.); junlongdai@link.cuhk.edu.hk (J.D.); yanliang@cuhk.edu.hk (L.Y.L.); wonglaihung@cuhk.edu.hk (G.L.-H.W.); wongv@cuhk.edu.hk (V.W.-S.W.); 2State Key Laboratory of Digestive Disease, Institute of Digestive Disease, The Chinese University of Hong Kong, Hong Kong SAR, China; 3Li Ka Shing Institute of Health Sciences, The Chinese University of Hong Kong, Hong Kong SAR, China

**Keywords:** cirrhosis, ascites, antiplatelet agents, anticoagulants, non-selective beta-blockers, albumin, diuretics, sodium-glucose cotransporter-2 inhibitors

## Abstract

Ascites is the most common complication from cirrhosis related to portal hypertension and depicts the onset of hepatic decompensation. Ranging from uncomplicated to refractory ascites, the progression carries prognostic value by reflecting the deterioration of underlying cirrhosis and portal hypertension. Diuretics have been the mainstay of treatment to control ascites, but the side effects heighten when the dosage is escalated. Non-selective beta-blockers (NSBBs) are widely used nowadays to prevent hepatic decompensation and variceal hemorrhage. However, with worsening systemic vasodilation and inflammation when ascites progresses, patients on NSBBs are at risk of hemodynamic collapse leading to renal hypoperfusion and thus hepatorenal syndrome. Long-term albumin infusion was studied to prevent the progression of ascites. However, the results were conflicting. Sodium-glucose cotransporter-2 inhibitors are under investigation to control refractory ascites. With that, patients with refractory ascites may require regular large-volume paracentesis. With an aging population, more patients are put on anti-thrombotic agents and their risks in decompensated cirrhosis and invasive procedures have to be considered. In general, decompensated cirrhosis with ascites poses multiple issues to pharmacological treatment. In the present review, we discuss the challenges and controversies in the pharmacological treatment of ascites.

## 1. Introduction

Decompensated cirrhosis represents the common final stage of chronic liver diseases [1]. The typical clinical course is characterized by compensated advanced chronic liver disease (cACLD), followed by decompensated cirrhosis with one or more complications such as ascites with or without spontaneous bacterial peritonitis (SBP), hepatic encephalopathy (HE), acute variceal hemorrhage (VH), jaundice and acute kidney injury (AKI) [2]. Decompensated cirrhosis represents a significant disease burden globally because of the high prevalence of different chronic liver diseases such as chronic viral hepatitis, alcohol-related liver disease, and metabolic dysfunction-associated steatotic liver disease (MASLD), leading to substantial morbidity and mortality with poor prognosis (Figure 1). The one-year mortality can be as high as 50% depending on the severity of liver dysfunction [3]. Ascites is the most common complication of decompensated cirrhosis, occurring in up to 60% of cirrhotic patients [3]. Development of ascites is associated with death in nearly 50% of the patients over two years if the underlying etiology of chronic liver disease is not properly managed [1].

Patients with decompensated cirrhosis often require frequent hospitalizations, notably for refractory ascites which warrants regular therapeutic paracentesis. That leads to a substantial economic burden with high direct medical costs and indirect costs due to lost productivity [4]. Both decompensated cirrhosis and ascites significantly impair patients’ quality of life, with symptoms such as abdominal distension, fatigue, and impaired physical functioning [1]. Patients with decompensated cirrhosis are at increased risk of developing other serious complications, such as SBP, hepatorenal syndrome (HRS), and hepatocellular carcinoma (HCC). Effective management strategies and early intervention are crucial to mitigate the significant impact of these complications on the patients and the healthcare system [4].

In this review, we undertook a literature search for articles published from January 2010 on MEDLINE and Web of Science using search terms “cirrhosis”, “ascites”, “non-selective beta-blockers”, “antiplatelets”, “anticoagulants”, “albumin”, and “sodium-glucose cotransporter 2 inhibitors”. Most relevant and impactful studies are discussed to illustrate various treatment options for patients suffering from mild to refractory ascites secondary to cirrhosis. Specifically, five controversies about antiplatelet agents and anticoagulants, non-selective beta-blockers (NSBBs), long-term albumin, diuretics and sodium-glucose cotransporter 2 inhibitors (SGLT2i) are discussed based on updated literature.

**Figure 1 pharmaceuticals-18-00339-f001:**
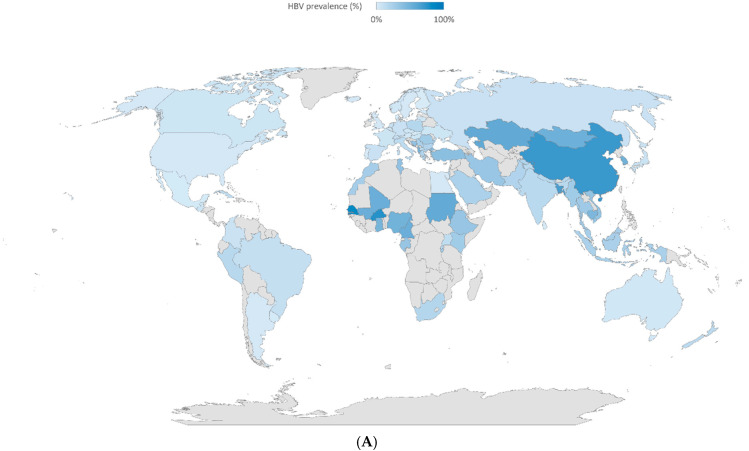
Global etiology of cirrhosis including (**A**). hepatitis B virus (HBV) infection, (**B**). hepatitis C virus (HCV) infection, (**C**). heavy alcohol use, and (**D**). metabolic dysfunction-associated steatotic liver disease (MASLD) in different regions. Data were obtained from a systemic review by Albert et al. which included studies on cirrhosis published between 1993 and 2021 [5].

## 2. Pathophysiology and Current Treatment Options for Ascites

### 2.1. Pathophysiology of Ascites

Ascites is a common manifestation of end-stage liver disease. In cirrhosis, due to the proliferation of intrahepatic fibrous tissue and nodular regeneration of hepatocytes, the liver undergoes structural remodeling leading to compression of branches of the portal vein and thus increased pressure in the portal venous system [6]. Concurrently, abnormal intrahepatic arterio-portovenous anastomoses form, with the inflow of arterial blood further exacerbating the increase in pressure. This pressure gradient is transmitted from the portal vein to the mesenteric veins, raising the hydrostatic pressure in terminal capillaries and causing fluid leakage into the abdominal cavity. Additionally, the remodeling of intrahepatic structures can compress or obstruct the hepatic veins, resulting in increased sinusoidal pressure. This increases lymph production but impairs effective lymphatic drainage, leading to lymph seeping from the liver surface, and contributing to the formation of ascites [7]. Furthermore, hepatic dysfunction leads to hypoalbuminemia which reduces plasma oncotic pressure that drives fluid into the interstitial space and eventually into the peritoneal cavity [8]. When a large volume of fluid accumulates in the portal venous system, reduced venous return to the heart decreases renal blood flow, leading to a reduction in glomerular filtration rate and decreased urine output [9]. Simultaneously, the renin–angiotensin–aldosterone system (RAAS) is activated, but impaired aldosterone inactivation due to liver dysfunction exacerbates sodium and water retention; Reduced urine output and circulating blood volume also stimulate antidiuretic hormone (ADH) secretion and inhibit atrial natriuretic peptide secretion, promoting tubular water reabsorption and resulting in dilutional hyponatremia [10]. Ultimately, these combined factors lead to the formation of ascites (Figure 2).

### 2.2. Clinical Implication of Ascites and Hyponatremia

Patients with ascites often report abdominal distension, discomfort, or pain. In cases of large-volume ascites, shifting dullness (detectable with 0.5–1.5 L of fluid) and a positive fluid wave (detectable with 1.5–2 L) are present on physical examination. Abdominal distension also thins the abdominal wall, increasing the risk of hernia formation. Additionally, diaphragmatic elevation from ascitic fluid can compress the heart and lungs, leading to shortness of breath.

Ascites marks the decompensated phase of cirrhosis. It may occur alone or be accompanied by other signs of portal hypertension such as HE and VH, which greatly complicate the management [11]. Importantly, ascites is associated with a poor prognosis, with studies indicating a significantly reduced five-year survival in cirrhotic patients with ascites. Patients with refractory ascites have an even worse prognosis, making it a key indication for liver transplantation. Additionally, ascites increases the risk of complications such as SBP, which, once developed, further worsens liver function with increased mortality [12].

Dilutional hyponatremia reflects fluid imbalance caused by liver disease and is an important marker of disease severity, closely linked to poor outcomes [13]. Persistent severe hyponatremia (i.e., serum sodium < 125 mmol/L) commonly occurs in patients with Child–Pugh C cirrhosis and may lead to HRS [14]. Hyponatremia presents additional risks in treatment, as diuretic use should be limited to avoid exacerbating electrolyte imbalance and renal dysfunction. This necessitates cautious diuretic titration and monitoring.

### 2.3. Treatment for Mild Ascites

The treatment of mild ascites begins with sodium and fluid restriction, forming the foundation of ascites management by reducing renal sodium and water reabsorption and limiting fluid accumulation [15]. Appropriate diuresis is also recommended, typically with potassium-sparing diuretics (e.g., spironolactone) or in combination with loop diuretics (e.g., furosemide), with dosage adjusted based on patient weight and serum electrolyte [16]. Subsequent management includes monitoring of fluid balance and maintaining homeostasis to prevent complications. In addition to symptomatic treatment, addressing the underlying cause of liver disease remains critical [11].

### 2.4. Treatment for Refractory Ascites (Figure 3)

Refractory ascites refers to ascites that cannot be mobilized or recurs after large-volume paracentesis (LVP) despite dietary sodium restriction and diuretic therapy [16,17]. Compared to regular paracentesis, LVP removes more than 5 L of ascitic fluid per session, but intravenous albumin supplementation is necessary to prevent paracentesis-related circulatory dysfunction which will lead to hypotension, hyponatremia, AKI and rapid reaccumulation of ascites [18]. While LVP provides immediate relief of symptoms, it does not prevent the recurrence of ascites. Another option is transjugular intrahepatic portosystemic shunt (TIPS) which creates a new portosystemic collateral by placing a stent between the portal and inferior vena cava to lower portal pressure and reduce ascites formation [19]. However, in patients with poor liver function, TIPS may lead to HE due to unfiltered venous blood with high ammonia level entering the systemic circulation. Other comorbidities, including poor cardiac function, also limit the use of TIPS. To address this, new techniques, such as the Alfapump^®^, are being explored [20]. The Alfapump^®^ is a subcutaneously implanted device that drains ascitic fluid into the bladder for natural excretion through urination. Preliminary clinical trials have shown promising results from the Alfapump^®^ for refractory ascites, though long-term follow-up studies are needed to further evaluate its safety and efficacy.

## 3. Controversy 1: Antiplatelet Agents and Anticoagulants in Patients with Ascites Who Need Paracentesis

Patients with cirrhosis suffer from a complex interplay of bleeding and thrombotic risks due to altered hemostatic balance [21]. Hemorrhagic tendency in the presence of coagulopathy (reflected by prolonged prothrombin time) and bleeding tendency (in the presence of thrombocytopenia) are apparent from the abnormal laboratory results. At the same time, these patients are also at heightened risk for thrombotic events. This paradoxical hypercoagulability arises from acquired deficiency of liver-synthesized protein C in cirrhosis combined with the preserved function of its co-factor, endothelial-derived thrombomodulin, resulting in a relatively hypercoagulable state [22]. VH is the commonest hemorrhagic event in patients with cirrhosis and is mostly pressure-driven by portal hypertension with little influence by hemostatic mechanisms [21]. Portal hypertension also contributes to the pathophysiology of cirrhotic portal vein thrombosis (PVT) as one pillar of the Virchow’s triad—with stasis of flow due to the same factors causing portal hypertension, together with the two other pillars of endothelial injury and hypercoagulability of cirrhosis [21].

Patients with cirrhosis are more often of advanced age with multiple comorbidities and are more likely to have indications for antiplatelet agents and anticoagulants. Common indications of antiplatelet agents are for cardiovascular comorbidities in which dual antiplatelet therapy (DAPT) with aspirin and clopidogrel is commonly used in patients with acute coronary syndrome or recent coronary stenting [23]. Common indications of anticoagulants include prophylaxis and/or treatment for thromboembolic events, such as cardioembolic stroke, deep vein thrombosis and pulmonary embolism [24].

The use of anticoagulants improves survival in patients with portal vein thrombosis due to underlying cirrhosis [25]. It has also been proposed that anticoagulants may reduce hepatic decompensation and improve survival in cirrhotic patients without an established indication for anticoagulation [26]. Given a postulation of microthrombi formation in cirrhosis which can be reversed by anticoagulation [27]. The field is gathering prospective data to verify whether anticoagulation can alter the natural history of cirrhosis, or which subgroup of patients with liver disease benefit most from anticoagulation. On the other hand, safety issues of antiplatelet agents and anticoagulants in patients with cirrhosis have drawn a lot of attention in view of their extensive use. Antiplatelet agents, including DAPT, are generally safe in patients with cirrhosis without increasing the incidence of major hemorrhagic events [23]. As hepatic dysfunction affects direct oral anticoagulants (DOACs) biotransformation to varying extents, some restrictions exist for the use of DOACs in patients with hepatic dysfunction based on the Child–Pugh grading (Table 1). Apixaban is the most reliant, up to 75%, on hepatic metabolism for drug elimination, followed by rivaroxaban (65%), edoxaban (50%), dabigatran (20%), and betrixaban (< 1%) [28,29]. DOACs, especially dabigatran, appear safe in patients with mild hepatic impairment (i.e., Child–Pugh A and/or B). However, all DOACs are contraindicated in patients with severe hepatic decompensation (i.e., Child–Pugh C) in which vitamin K antagonist (VKA) (i.e., warfarin) is the only recommended anticoagulant in this population [30]. There was no significant difference between DOAC and VKA users in rates of procedure-related bleeding (within 4 weeks of banding, paracentesis, liver biopsy, or surgery) and spontaneous bleeding events [23]. While there is no clear guideline to suggest the duration of drug interruption, paracentesis per se is considered a low-risk procedure with <1.5% bleeding risk [31]. Platelet count and clotting profile are not required before paracentesis except in warfarin users to avoid over-warfarinization [31,32]. In general, antiplatelet agents and DOACs can be continued for paracentesis, which is similar to other low-risk procedures such as thoracocentesis for hepatic hydrothorax [32]. By the same token, endoscopic procedures of low-risk of bleeding, such as diagnostic endoscopy with mucosal biopsy, endoscopic ultrasound without fine needle aspiration or biopsy, and gastrointestinal luminal stenting, can be performed without interruption of antiplatelet agents or DOACs. On the other hand, endoscopic therapies on varices including endoscopic banding ligation and injection of glue or sclerosants, which are common in patients with cirrhosis, are considered procedures with high-risk of bleeding [33]. In the setting of elective procedures, it is recommended that clopidogrel (and ticagrelor/prasugrel), and anticoagulants should be withheld prior to procedures with high bleeding risk, while aspirin can be continued. Thienopyridines (such as clopidogrel) should be withheld for 5 days before procedure. Factor Xa inhibitors such as apixaban, edoxaban or rivaroxaban, should be withheld at least 2 days before the procedure. While for dabigatran, in which the metabolism is reliant on kidney function, needs to be withheld 2 days before the procedure if the creatinine clearance (CrCl) is >80 mL/min, 3 days if CrCl is between 50 and 80 mL/min and 4 days if that is between 30 and 50 mL/min. Warfarin, otherwise, has to be withheld with monitoring of the international normalized ratio (INR) aiming at 1.5–1.8 or lower before procedure [32,33]. Of note, these recommendations are mostly based on retrospective observational studies and the definition on the procedural bleeding risk varies across guidelines. More prospective data and standardized recommendations are needed to ascertain whether there is a need to stop any of these agents, and for how long these agents should be stopped, for various procedures commonly performed on patients with cirrhosis.

**Table 1 pharmaceuticals-18-00339-t001:** Recommendations on direct oral anticoagulants (DOACs) based on degree of hepatic decompensation [30]. Table 2 shows the calculation of Child–Pugh score.

	Hepatic Metabolism (%)	Half-Life (h)	Safe to Use Without Dose Reduction	Use with Caution	Contraindication
Apixaban	75	12	Child–Pugh A	Child–Pugh B	Child–Pugh C
Betrixaban	<1	19–27	Child–Pugh A	Not applicable	Child–Pugh B or C
Dabigatran	20	12–17	Child–Pugh A	Child–Pugh B	Child–Pugh C
Edoxaban	50	10–14	Child–Pugh A	Child–Pugh B	Child–Pugh C
Rivaroxaban	65	7–13	Child–Pugh A	Not applicable	Child–Pugh B or C

**Table 2 pharmaceuticals-18-00339-t002:** Calculation of Child–Pugh score. Child–Pugh score is calculated by summing the points for all five parameters. Class A, B, and C correspond to 5–6 points, 7–9 points, and 10–15 points, respectively.

Parameter	1 Point	2 Points	3 Points
Total bilirubin	<2 mg/dL (<34 μmol/L)	2–3 mg/dL (34–50 μmol/L)	>3 mg/dL (>50 μmol/L)
Serum albumin	>3.5 g/dL (>35 g/L)	2.8–3.5 g/dL (28–35 g/L)	<2.8 g/dL (<28 g/L)
PT prolongation (s) or INR	<4 s or INR < 1.7	4–6 s or INR 1.7–2.3	>6 s or INR > 2.3
Ascites	None	Mild	Moderate to severe
Hepatic encephalopathy	None	Grade 1–2	Grade 3–4

## 4. Controversy 2: Non-Selective Beta-Blockers

NSBBs, such as propranolol and nadolol, block both beta-1 and beta-2 adrenergic receptors to reduce the cardiac output and splanchnic blood flow. As a result, portal pressure is reduced. Carvedilol, with extra alpha-1 adrenergic receptor blockade, further reduces the portal pressure by 7–8% more than propranolol as it lowers the intrahepatic vascular resistance [34,35]. Thus, carvedilol is currently the beta-blocker of choice to treat portal hypertension [36,37,38].

Conventionally, NSBBs are initiated to prevent the first occurrence (i.e., primary prophylaxis) or recurrence (i.e., secondary prophylaxis) of VH when there are gastroesophageal varices, regardless of the status of cirrhosis. Most data came from esophageal varices (EV) in which NSBBs are used alone as primary prophylaxis or in combination with endoscopic variceal ligation as secondary prophylaxis [37,39]. Similarly, patients with gastric varices or portal hypertension gastropathy should be started on NSBBs to reduce the risk of hemorrhage [40].

It was not an indication to start NSBBs in patients without gastroesophageal varices or ascites until the PREDESCI trial shed light on the use of NSBBs to prevent decompensation in patients with compensated cirrhosis. The trial showed that in patients with compensated cirrhosis and clinically significant portal hypertension (CSPH) based on hepatic venous pressure gradient (HVPG) ≥10 mmHg, the use of NSBBs led to an 11% absolute risk reduction in hepatic decompensation (16% vs. 27% for those taking placebo; hazard ratio [HR] 0.51, 95% confidence interval [CI] 0.26–0.97; *p* = 0.041), mainly by reduction in incident ascites [41]. Together with the ANTICIPATE model suggesting the combination of liver stiffness measurement (LSM) from vibration-controlled transient elastography (VCTE) and platelet count in predicting CSPH [42], the management of patients cACLD has faced a paradigm shift. For patients with cACLD and a high likelihood of CSPH by LSM and platelet criteria (i.e., LSM ≥ 25 kPa, or LSM 20–25 kPa with platelet count <150 × 10^9^/L), NSBBs (in particular, carvedilol) can be initiated to prevent hepatic decompensation [36,37]. This stresses the potential role of VCTE as a diagnostic tool with prognostic value that may also serve as a monitoring modality. However, it is noteworthy that there were no prospective data to validate the noninvasive assessment-based approach in NSBBs or carvedilol initiation. With that, at least two ongoing multicenter trials (ChiCTR2300073864 and NCT06449339) are exploring the benefits of carvedilol based on LSM by VCTE and platelet count alone [43].

NSBBs reduce portal pressure to reduce the risk of VH. On the other hand, patients in decompensation have fragile hemodynamic status owing to the underlying pathophysiology with systemic vasodilation and inflammation [16,44]. A further drop in systemic blood pressure does not only lead to hypotension-related symptoms, but may also potentiate hemodynamic collapse, HRS and affect survival [45]. In particular, there were safety concerns about the use of NSBB in patients with decompensated cirrhosis and refractory ascites due to a higher risk of paracentesis-related circulatory dysfunction and mortality. A single-center prospective observational study with 151 patients showed a significant reduction in median survival (5.0 months vs. 20.0 months, *p* < 0.001) for those who received propranolol (mean dosage 113 mg/day) to prevent VH compared to those who did not receive the drug, respectively [46]. Another retrospective study suggested that the use of NSBBs in patients with SBP increased the risk of HRS and hemodynamic collapse, and worsened transplant-free survival [47]. Conflicting results otherwise suggested the opposite by showing that NSBBs had no impact or even improvement on mortality and development of HRS [48]. For instance, a study showed improved mortality in patients with ascites listed for liver transplantation receiving NSBBs compared to those who did not receive NSBBs (adjusted HR 0.55, 95% CI 0.32–0.95, *p* = 0.032). The improvement in survival remained similar in those with refractory ascites [49]. One of the postulations pointed towards the potential effect of NSBBs on reducing systemic inflammatory response which plays an important role in the overall pathophysiology of disease progression [50,51]. With the encouraging result from the PREDESCI trial to show the reduced risk of ascites in patients receiving NSBBs, a window period hypothesis was raised to propose the use of NSBBs only in a certain stage of cirrhosis.

Whilst there was debatable evidence, the use of NSBBs with optimal dosage did not appear to aggravate the risk on survival even in decompensated cirrhosis [52]. It was shown that the optimal dosage for propranolol should be capped at 160 mg daily and be titrated against blood pressure and heart rate, while that of carvedilol should be maintained at 6.25–12.5 mg daily, for patients with ascites [37]. A supra-therapeutic dosage does not likely confer additional reduction in portal pressure but an increased risk of side effects from the NSBBs [34,40]. Hence, careful titration to maintain a NSBB dosage within the desired range should be advocated while further studies on the safety and efficacy of NSBBs in patients with decompensated cirrhosis, especially with refractory ascites, are awaited. Table 3 summarizes the risks and benefits of NSBBs at different stages of cirrhosis.

## 5. Controversy 3: Long-Term Albumin

Albumin possesses the ability to maintain intravascular volume by increasing the oncotic pressure and is indicated for LVP to prevent paracentesis-related circulatory dysfunction, and for SBP and HRS to preserve the renal function. The duration of albumin infusion for these indications is short in terms of days. Emerging evidence has suggested another property of albumin in which it may bind to ligands and endotoxin. Its anti-inflammatory and anti-oxidant effects mitigate the inflammatory response in cirrhosis and liver failure [16,53,54]. With that, several studies have investigated whether long-term albumin infusion would improve the control of ascites and as well alter the natural course of underlying cirrhosis. The landmark ANSWER trial randomized 431 patients with diuretic-responsive ascites to receive either standard medical treatment or in combination with 40 g of albumin twice a week for the first 2 weeks and then 40 g weekly for 18 months. The overall 18-month survival was significantly higher in the combination arm than the standard medical treatment group arm (Kaplan–Meier estimates 77% vs. 66%, respectively, *p* = 0.028) which conferred to a 38% reduction in mortality HR. Notably, nearly twice the number of patients in the combination arm remained free of paracentesis throughout the study period compared to the standard medical treatment arm [55]. On the contrary, the MACHT trial recruited 173 patients with ascites listed for liver transplantation and randomized them into 40 g albumin every 15 days with midodrine, or placebo, on top of standard diuretic treatment. Despite some favorable signals for the improvement of circulatory dysfunction, the albumin-midodrine regimen did not result in a significant improvement in survival or prevention of cirrhotic complications [56]. The differences in findings between the two trials could be explained by different study populations. First, the participants in the ANSWER trial were more stable with uncomplicated ascites, whereas those in the MACHT trial were listed for liver transplantation with a higher Model for End-stage Liver Disease (MELD) score. The mean duration of albumin given was longer in the ANSWER trial compared to that in the MACHT trial (14.5 months vs. 8 days, respectively). Similarly, the dosage of albumin given in the ANSWER trial was higher [57]. The discrepant results were further complicated by another prospective trial in Italy which included patients with cirrhosis and refractory ascites to receive long-term albumin at 20 g twice per week to show a significantly lower 24-month mortality and emergent hospitalizations compared to those who did not receive long-term albumin [58]. The complex interplay among different severity of ascites and cirrhosis, and the dosage and duration of long-term albumin added uncertainty to the true effect of this treatment on patients with cirrhosis and ascites. This was reinforced by a recent systematic review and meta-analysis involving five randomized controlled trials to suggest an insignificant difference on the use of long-term albumin in improving mortality or other complications of cirrhosis compared to the control groups [59].

Apart from the inconsistency in therapeutic effects of long-term albumin infusion, there was also debate on whether there is a magic figure for serum albumin that one should aim for while treating patients with ascites. The post hoc analysis of ANSWER trial suggested a potential target of serum albumin 40 g/L at 1-month of long-term albumin treatment regardless of the baseline serum albumin level as the albumin level at this time point predicted clinical outcomes [60]. However, the ATTIRE trial which studied on the short-term use of albumin to increase the serum albumin level to a target of 30 g/l in hospitalized patient with decompensation cirrhosis failed to show superiority to standard treatment in terms of short-term risk of infection, kidney dysfunction and death. Meanwhile, the risk of fluid overload and pulmonary edema was higher in those who received albumin treatment [61]. Although the study designs and populations were different across studies, and there have not been apparent negative safety signals arising from long-term albumin, the potential risk of fluid imbalance has to be carefully monitored. Table 4 summarizes the potential benefits and risks of long-term albumin at different stages of cirrhosis.

## 6. Controversy 4: Diuretics

Diuretics are primarily used in the treatment of conditions such as hypertension, heart failure and renal insufficiency. There are similar pathophysiological mechanisms between cirrhosis and heart failure in which splanchnic vasodilation in cirrhosis and reduced cardiac output in heart failure both lead to a reduction in effective circulatory volume. Both conditions consequently activate the RAAS. Spironolactone competitively inhibits aldosterone receptors, thereby diminishing the effectiveness of aldosterone and decreasing the reabsorption of sodium and free water from the kidneys. Hyperaldosteronism worsens sodium retention and leads to resistance to loop diuretics, which makes aldosterone antagonists work better than loop diuretics. This underscores the primary role of aldosterone antagonists in the management of moderate ascites [62]. Furosemide is a short-acting loop diuretic which inhibits the sodium-potassium-chloride cotransporter 2. This prevents the reabsorption of sodium and chloride in the ascending limb of the loop of Henle. A randomized clinical trial evaluated the efficacy and safety of sequential versus combined diuretic therapy in non-azotemic patients with cirrhosis and moderate ascites. The study found that combined therapy, which involved concurrent use of aldosterone antagonists and loop diuretics, was superior for its faster and more effective control of ascites. Furthermore, this approach resulted in fewer side effects compared to sequential therapy, where loop diuretics were added after aldosterone antagonists [63]. Several complications including AKI, electrolyte disturbance, HE, gynecomastia, and muscle cramps are associated with diuretics use. Close monitoring of these patients is essential, especially during the first month, to detect and manage these issues in a timely manner.

In many clinical guidelines, the treatment for moderate ascites begins with aldosterone antagonists. However, the starting and maximum dosages of these drugs vary globally. The European and American guidelines suggest similar use of diuretics in patients with a first episode of grade 2 (moderate) ascites [15,16,64]. The recommended initial treatment involves spironolactone starting at 100 mg/day and potentially varying between 100 and 200 mg/day. The dose can be increased in increments of 100 mg every 72 h up to a maximum of 400 mg/day if no response is observed. If the patient does not respond or develops hyperkalemia, furosemide is added with a starting dose of 40 mg/day and increasing to a maximum of 160 mg/day in 40 mg increments [15,16]. For patients with long-standing or recurrent ascites, a combination therapy with spironolactone and furosemide is used with dosage adjustments based on response. The effectiveness of diuretics is monitored by daily body weight reduction to achieve a negative sodium balance. However, the Japanese clinical guidelines recommended lower doses of diuretics for patients with cirrhosis due to a higher sensitivity to adverse effects [65,66]. For both Grade 2 (moderate) and Grade 3 (large) ascites, spironolactone is prescribed as the first-line treatment, starting at 25 to 50 mg/day and increasing to a maximum of 100 mg/day if necessary. If spironolactone monotherapy is proven ineffective, a loop diuretic (i.e., furosemide) is added with a starting dosage of 20 to 40 mg/day and potentially increasing up to 80 mg/day based on response.

Vaptans are selective antagonists of vasopressin receptor 2 located in the principal cells of the collecting ducts. They function by inhibiting the interaction between ADH and these receptors, promoting the excretion of free water by altering sodium levels in the urine. Tolvaptan is a new type of vaptans and has been approved in America and Europe for treating severe hypervolemic hyponatremia, syndrome of inappropriate antidiuretic hormone secretion (SIADH), heart failure and liver cirrhosis. A phase 3 multicenter clinical trial investigated the efficacy of tolvaptan as an add-on therapy in liver cirrhosis patients who showed insufficient response to conventional diuretics. The study found that tolvaptan could reduce body weight significantly and improve serum sodium concentration compared to placebo, suggesting an effective clearance of solute-free water. With the improvement of liver cirrhosis-related complications, tolvaptan seems to be a valuable add-on therapy for managing patients with cirrhosis, offering improvements in clinical symptoms and biochemical profiles for short durations [67]. However, tolvaptan showed hepatotoxic effects in patients with autosomal dominant polycystic kidney disease which typically appears between 3 and 18 months after starting the medication. The study emphasizes the necessity of monitoring liver function and careful use of vaptans in patients with cirrhosis after weighing the potential risks and benefits [68]. Another study demonstrated the safety and effectiveness of tolvaptan in managing ascites in patients with decompensated liver cirrhosis without severe adverse effects on liver or renal function over 6 months [69]. With the potential risk, the United States Food and Drug Administration (FDA) issued a warning that tolvaptan should be not used for longer than 30 days and be used in patients with underlying liver disease owing to the possibility of liver injury and potential liver transplantation and death. Apart from Japan where tolvaptan is used as a second-line therapy for ascites when traditional diuretics fail and with favorable renal function [65,66], the optimal duration for tolvaptan therapy remains undetermined and the agent is not routinely recommended for treating ascites from cirrhosis.

## 7. Controversy 5: Sodium-Glucose Co-Transporter-2 Inhibitors

SGLT2i belongs to a novel class of glucose-lowering medications, which works by causing glycosuria, making them an appealing option for patients with cirrhosis [70]. SGLT2i is well proven to improve glycemic control in patients with type 2 diabetes (T2D) as well as metabolic liver disease, particularly MASLD [70]. Beyond the antidiabetic effects, SGLT2i has cardioprotective benefits which expands its potential use to heart failure patients without T2D [71]. Since decompensated cirrhosis and heart failure share some common pathophysiological features, SGLT2i may also benefit patients with decompensated cirrhosis, even in the absence of T2D [70].

SGLT2i was originally designed for glycemic control in T2D, but early clinical evaluations uncovered their significant impact on major adverse cardiovascular events and renal dysfunction [72]. Given multiple major clinical benefits with SGLT2i, namely improved glycemic control with minimal hypoglycemia, weight reduction, reduced heart failure hospitalizations, reduced albuminuria, and lowered blood pressure and cholesterol levels, the approved indications of SGLT2i have been expanded from T2D only to patients with established cardiovascular disease, heart failure with reduced or preserved ejection fraction, and chronic kidney disease (CKD) [72]. On top of the approved indications, MASLD is a well-recognized off-label indication for SGLT2i, as the American Association of Clinical Endocrinology (AACE) recommends SGLT2i as adjunctive therapy in patients with T2D and MASLD [73].

Shared pathophysiological alternations between congestive heart failure and decompensated cirrhosis include arterial under-filling and hence compensatory activation of the RAAS, sympathetic nervous system (SNS) and ADH secretion [70]. The common net effect of these mechanisms is sodium and water retention, expansion of extravascular fluid volume, and consequently ascites and peripheral edema. On the other hand, systemic inflammation combined with immune dysfunction and severe hypoalbuminemia plays a more important role in cirrhosis than in heart failure [74].

There is currently limited data available on the use of SGLT2i in patients with decompensated cirrhosis. A handful of case reports have been published over the last few years, which described some patients who experienced weight loss, improved fluid retention, and resolution of ascites and saving paracentesis without electrolyte disturbances with SGLT2i [70]. A more recent cohort study adopted a propensity-score matching analysis of patients with T2D and cirrhosis demonstrated a lower rate of incident ascites (3.8% vs. 7.1%) and death (0.9% vs. 4.5%) in SGLT2i users compared to those who used dipeptidyl peptidase-4 inhibitors (DPP4i), although the difference in ascites rates was not statistically significant. The apparently favorable role of SGLT2i in patients with decompensated cirrhosis warrants further investigation [75].

Using SGLT2i in patients with decompensated cirrhosis is not without risks: these patients are at increased risk of infection, which further increases the risk of genitourinary infections due to SGLT2i; patients who have a relative depletion of effective circulatory volume may rapidly develop AKI secondary to volume depletion due to SGLT2i-induced glycosuria [76]. Euglycemic diabetic ketoacidosis is also an important concern. Therefore, risk-benefit evaluation is necessary before its use in patients with cirrhosis.

## 8. Limitations

We conducted a narrative review rather than a systematic review, to synthesize the most recent evidence on the challenges and controversies related to the pharmacological management of ascites. This review draws on expert perspectives and focuses on selected key studies in the field. Consequently, the article selection process was informed by the authors’ expertise and their understanding of the most significant and influential research on the subject.

## 9. Future Directions

Emerging data suggest that it is safe to continue antiplatelet drugs and anticoagulants during paracentesis, and safety data will likely accumulate in future real-world cohorts. However, similarly to other medical managements of decompensated cirrhosis, robust and widely available tests for coagulation to guide the safety of invasive procedures and administration of anticoagulants remain unmet clinical needs [77]. The PREDESCI trial suggests that patients with CSPH would benefit from NSBBs or carvedilol even in the absence of high-risk varices, but the measurement of HVPG is impractical in routine practice [41]. Rather, at least two ongoing trials (NCT06449339 and Liu et al.) are exploring the benefits of carvedilol based on noninvasive assessments using LSM by VCTE and platelet count alone [43].

Albumin infusion is a time-honored treatment during LVP and in patients with HRS-AKI and hypotension. Despite several well-conducted randomized controlled trials, the benefits of regular maintenance albumin infusion remain uncertain [55,56]. It is likely that the benefit is restricted to specific clinical contexts. Future studies should harmonize clinical trial design and the definition of trial endpoints.

Finally, SGLT2i has demonstrated beneficial effects on glycemic control, weight reduction, kidney function, and heart failure. Many of these favorable physiological changes are relevant to patients with decompensated cirrhosis [76]. However, there have been few dedicated clinical trials in this high-risk population. In addition, it is important to scrutinize the safety of SGLT2i in patients with cirrhosis in large real-world cohorts, especially regarding genitourinary infection, euglycemic diabetic ketoacidosis, and AKI.

## 10. Conclusions

Ascites, the most common cirrhotic complication, affects nearly 60% of individuals with cirrhosis and significantly influences both patient quality of life and prognosis. Ascites does not only impair physical functioning but also heightens the risk of developing serious complications like SBP and HRS. Controlling the underlying etiological factors alongside regular monitoring with timely detection, medical interventions and pharmacological treatment for ascites are pivotal in mitigating the impact of these complications on patients and the healthcare system. This review summarizes the emerging pharmacological approaches for treating ascites in different stages of cirrhosis, encompassing antiplatelet agents, anticoagulants, NSBBs, long-term albumin infusion, and SGLT2i, and highlights the knowledge gaps.

## Figures and Tables

**Figure 2 pharmaceuticals-18-00339-f002:**
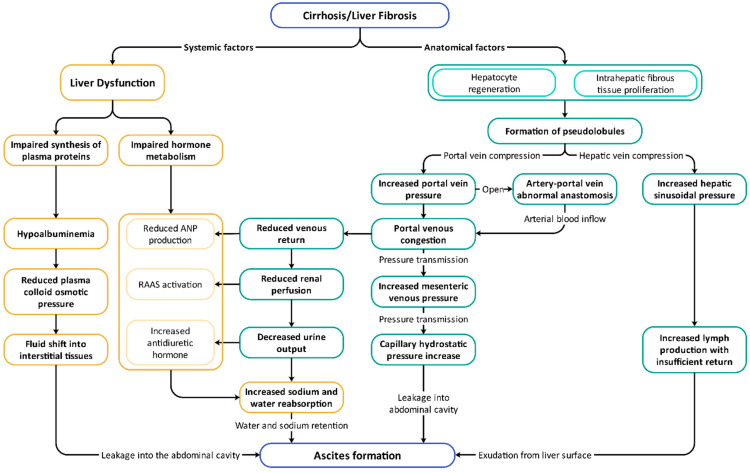
Pathophysiology of ascites in cirrhosis.

**Figure 3 pharmaceuticals-18-00339-f003:**
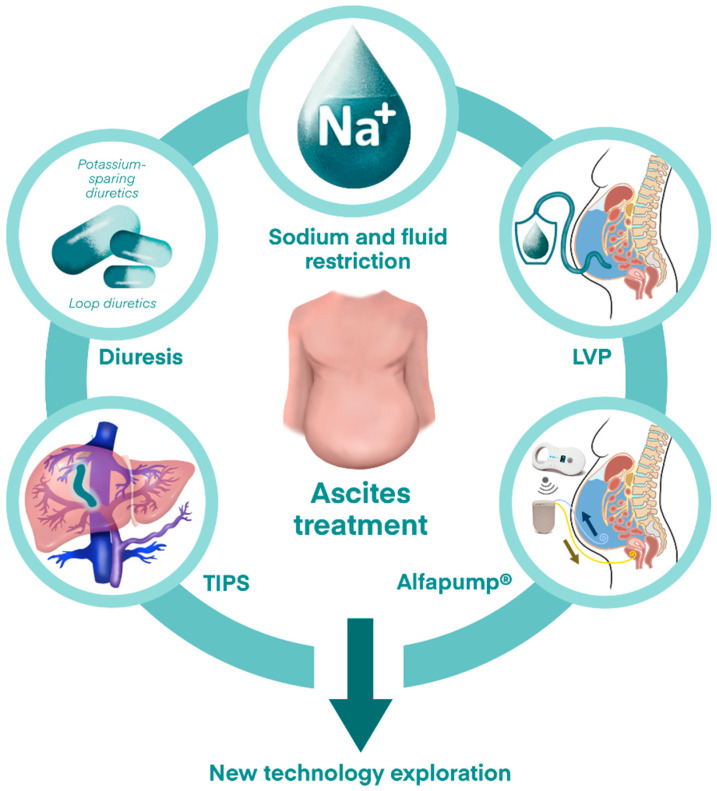
Treatment options for refractory ascites.

**Table 3 pharmaceuticals-18-00339-t003:** Risks and benefits of non-selective beta-blockers (NSBBs) at different stages of cirrhosis.

	cACLD Without CSPH	cACLD with CSPH	Decompensated Cirrhosis (with Uncomplicated Ascites)	Decompensated Cirrhosis (with Refractory Ascites)
Benefits	- No proven benefit	- Prevent incident hepatic decompensation and improve transplant-free survival - Prevent variceal hemorrhage	- Prevent variceal hemorrhage	- Prevent variceal hemorrhage - Debatable evidence on improvement in survival
Risks	- NSBBs-related side effects	- NSBBs-related side effects	- NSBBs-related side effects, especially hypotension	- NSBBs-related side effects, especially hypotension - Debatable risk of hepatorenal syndrome and mortality

cACLD = compensated advanced chronic liver disease, CSPH = clinically significant portal hypertension. cACLD includes patients with advanced liver fibrosis or compensated cirrhosis diagnosed clinically, histologically or by noninvasive assessments (such as vibration-controlled transient elastography (VCTE). CSPH is diagnosed by hepatic venous pressure gradient (HVPG) ≥ 10 mmHg, liver stiffness measurement (LSM) ≥ 25 kPa on certain chronic liver disease etiologies per Baveno VII consensus [36] or the presence of varices on esophagogastroduodenoscopy.

**Table 4 pharmaceuticals-18-00339-t004:** Potential benefits and risks of long-term albumin at different stages of cirrhosis.

	Uncomplicated Ascites	Refractory Ascites	Decompensated Cirrhosis Listed for Transplantation
Type of study	- Randomized, open-label, controlled trial (ANSWER trial) [55]	- Non-randomized trial [58]	- Randomized, double-blind, placebo-controlled trial (MACHT trial) [56]
Number and characteristics of study population	- 431 patients - With cirrhosis and uncomplicated ascites who were treated with anti-aldosteronic drugs (≥200 mg/day) and furosemide (≥25 mg/day) (standard medical therapy)	- 70 patients - With cirrhosis and refractory ascites	- 173 patients - With cirrhosis and ascites awaiting liver transplantation
Study groups	- Standard medical therapy plus albumin infusion (n = 218) vs. standard medical therapy (n = 213)	- Standard medical therapy plus albumin infusion (n = 45) vs. standard medical therapy (n = 25)	- Midodrine (15–30 mg/day) plus albumin infusion (n = 87) vs. matching placebos (n = 86)
Albumin infusion regime	- 40 g of albumin twice a week for the first 2 weeks and then 40 g weekly for up to 18 months	- 20 g of albumin twice a week	- 40 g of albumin infusion every 15 days up to 1 year, until liver transplantation or drop-off from inclusion on waiting list
Benefits	- Improved 18-month survival (77% vs. 66%, *p* = 0.028) - Reduction in need of paracentesis	- Lower 24-month mortality (41.6% vs. 65.5%, *p* = 0.032) - Lower emergent hospitalizations	- Did not improve survival (*p* = 0.527) or prevent cirrhotic complications (*p* = 0.402)
Risks	- Comparable grade 3–4 non-liver related adverse events	- No added safety concerns	- No significant differences in adverse events between the two groups

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
