# Peer review of "Pharmacological Treatment of Ascites: Challenges and Controversies"

_pharmaceuticals, 2025, doi:10.3390/ph18030339_

Round 1
Reviewer 1 Report
Comments and Suggestions for Authors
This is a nice review regarding the challenges and controversies in the treatment of ascites and in particular the use of diuretics, NSBBs, albumin infusions and SLGT2i. They also discuss the use of anti-coagulation therapy with emphasis to DOACS in this population.
Commnets/suggestions
A. CONTROVERSY 1
-The hepatic metabolism of DOACS along with their half life to be included in Table 1
- it would be more informative to include in the text which procedures are of low risk for bleeding and that no drug interruption is needed
- in the same line for major procedures which is the suggested time of drug interruption, depending on the anti-coagulation drug used, should be mentioned
A comment for the possible role of anticoagulation drugs in the natual history of liver disease should be also mentioned
B CONTROVERSY 3 Albumin infusions
- table 3 should be expanded with more details regarding the studies (type of study, number and characteristics of patients , protocol of infusion, efficacy/benefits and side effects
Author Response
Please kindly see attached the point-by-point response.

Reviewer 2 Report
Comments and Suggestions for Authors
Your review summarizes the emerging pharmacological approaches for treating ascites. Have you evaluated cirrhosis in patients without clinically obvious ascites which could lead to better outcomes?
Author Response
Please find attached the point-by-point response.

Reviewer 3 Report
Comments and Suggestions for Authors
The authors presented a review regarding Pharmacological treatment of ascites and different therapeutic approaches. Here are some recommendations for the manuscript:
Abbreviations: once introduced abbreviation, there is no need for another introduction. For instance, NSBBs is introduced in Line 58, then again in Line 190. Please, check the whole manuscript and make adjustments if necessary.
Abstract: I suggest to the authors some minor changes:
Line 12, instead word commonest-most common.
Please, change the word domprognostic, Line 14.
NSBBs – introduce abbreviation.
Introduction:
Line 46 – word That, replace the capital letter.
Pathophysiology and current treatment options for ascites:
Figure 2. Since all starts with capital letters, hepatocyte regeneration also should be capitalized.
Treatment for refractory ascites:
The authors could add a new figure/scheme regarding this subsection.
Clinical implication of ascites and hyponatremia:
Line 100, please check the font size
Table 1. Could authors add an explanation for Child‐Pugh grading? Maybe as an additional table?
Table 2. Please, make graphical adjustments (Justify, etc.)
The authors should add a section dedicated to the research strategy of studies used in this review.
Controversy 3:
Line 274: word Nearly-small letter
Line 301: 40 g/L – capital L
The authors could add limitations to their study.
Conclusion:
One paragraph regarding the main conclusion at the end should be added.
Author Response

(The authors gave the same response as above.)

Round 2
Reviewer 3 Report
Comments and Suggestions for Authors
The authors have improved the manuscript, and it is suitable for publication.
Author Response
Attached please find the point-by-point response to the Reviewers' comments. Thank you very much.
